# A Novel One-Pot Synthesis and Characterization of Silk Fibroin/α-Calcium Sulfate Hemihydrate for Bone Regeneration

**DOI:** 10.3390/polym13121996

**Published:** 2021-06-18

**Authors:** Aditi Pandey, Tzu-Sen Yang, Shu-Lien Cheng, Ching-Shuan Huang, Agnese Brangule, Aivaras Kareiva, Jen-Chang Yang

**Affiliations:** 1Graduate Institute of Nanomedicine and Medical Engineering, College of Biomedical Engineering, Taipei Medical University, Taipei 11052, Taiwan; aditi8293@tmu.edu.tw (A.P.); lillian704372002@yahoo.com.tw (S.-L.C.); 2Graduate Institute of Biomedical Optomechatronics, Taipei Medical University, Taipei 11031, Taiwan; tsyang@tmu.edu.tw; 3School of Dentistry, College of Oral Medicine, Taipei Medical University, Taipei 11031, Taiwan; jollyhuangtw12@gmail.com; 4Department of Pharmaceutical Chemistry, Riga Stradins University, LV-1007 Rīga, Latvia; Agnese.Brangule@rsu.lv; 5Institute of Chemistry, Vilnius University, Naugarduko 24, LT-03225 Vilnius, Lithuania; aivaras.kareiva@chgf.vu.lt; 6International Ph.D. Program in Biomedical Engineering, College of Biomedical Engineering, Taipei Medical University, Taipei 11031, Taiwan; 7Research Center of Biomedical Device, Taipei Medical University, Taipei 11052, Taiwan; 8Research Center of Digital Oral Science and Technology, Taipei Medical University, Taipei 11052, Taiwan

**Keywords:** silk fibroin, calcium sulfate hemihydrate

## Abstract

This study aims to fabricate silk fibroin/calcium sulfate (SF/CS) composites by one-pot synthesis for bone regeneration applications. The SF was harvested from degummed silkworm cocoons, dissolved in a solvent system comprising of calcium chloride:ethanol:water (1:2:8), and then mixed with a stoichiometric amount of sodium sulfate to prepare various SF/CS composites. The crystal pattern, glass transition temperature, and chemical composition of SF/CS samples were analyzed by XRD, DSC, and FTIR, respectively. These characterizations revealed the successful synthesis of pure calcium sulfate dihydrate (CSD) and calcium sulfate hemihydrate (CSH) when it was combined with SF. The thermal analysis through DSC indicated molecular-level interaction between the SF and CS. The FTIR deconvolution spectra demonstrated an increment in the β-sheet content by increasing CS content in the composites. The investigation into the morphology of the composites using SEM revealed the formation of plate-like dihydrate in the pure CS sample, while rod-like structures of α-CSH surrounded by SF in the composites were observed. The compressive strength of the hydrated 10 and 20% SF-incorporated CSH composites portrayed more than a twofold enhancement (statistically significant) in comparison to that of the pure CS samples. Reduced compressive strength was observed upon further increasing the SF content, possibly due to SF agglomeration that restricted its uniform distribution. Therefore, the one-pot synthesized SF/CS composites demonstrated suitable chemical, thermal, and morphological properties. However, additional biological analysis of its potential use as bone substitutes is required.

## 1. Introduction

The field of restoration and regeneration of critical bone defects, although challenging for scientists, has amazed humanity by its advancements through the changing times [1]. This trend of exploring new methods in the context of bone defects assists in developing techniques for accelerating bone healing. The age-old practice of bone grafting for replacing lost bone involves osteogenesis, osteoconduction, and osteoinduction, which promotes regeneration at the affected site [2]. However, complications, such as the need for a second surgery (in autografting), and immunogenicity (in allografts and xenografts) led to the development of synthetic bone substitutes (alloplasts) [3,4].

Calcium sulfate (CS) has three crystalline phases, namely, anhydrite (CaSO_4_), hemihydrate (CaSO_4_·0.5H_2_O), and dihydrate (CaSO_4_·2H_2_O). Upon mixing the calcium sulfate hemihydrate (CSH) with water, the calcium sulfate dihydrate (CSD) is formed exothermically. Calcium sulfate hemihydrate compounds are commonly used as bone substitutes due to their biocompatibility, resemblance with bone mineral, excellent osteoinduction property, supportive scaffolds for osseous regeneration (delivering growth factors), and potential cost-effectiveness [4,5,6,7]. It is important to control the shape and size of CS, as its performance is related to such applications [8,9,10]. There are two distinct hemihydrate forms, α and β, among which, α-calcium sulfate hemihydrate (α-CSH) has been used more extensively for medical applications due to its high strength [11,12]. The α-CSH form usually dissolves slower in comparison to the β-CSH form due to its density and stability; thus, it is often chosen for bone-filling applications [13].

The main limitation of calcium sulfate bone substitutes is its fast resorption, which exceeds the bone formation/growth rate at the defect site, leading to the creation of pores after its degradation [14,15,16]. Furthermore, despite the bone regeneration property of CS bone cements, they need to be combined with another tougher material to achieve enhanced mechanical performance [17]. Improving strength and reducing the resorption rate of CS could be attained by the addition of organic or polymeric materials. As reported, polymeric fibers, in combination with bioactive components, depicted cell attachment and differentiation. The ideal polymeric materials must promote the formation and maintenance of controlled matrix heterogeneity, along with morphological and material properties resembling the native insertion site [18]. A biodegradable polymer, such as collagen, which is a natural bone-ingredient, was blended with CS and improved its biological activity, but enhancement of mechanical strength was not reported [19].

Silk fibroin (SF), a protein harvested from the cocoons of the *Bombyx mori* silkworm, comprises of a 26 kDa light chain and a 391 kDa heavy chain linked by a single disulfide bond, forming an H–L complex [20]. SF is a promising biomaterial, which is used as films, microspheres, fibers, and porous scaffolds [21] for biomedical applications, such as drug delivery, wound healing, and bone tissue engineering [22,23,24,25]. Its characteristics, including excellent biocompatibility and mechanical properties, provide it with the potential ability to support the matrix for fibroblasts, osteoblasts and hepatocytes, and applications in ligament tissue engineering [26,27]. Mieszawska et al. reported that, SF, when combined with a silica-based material, exhibited osteoinductive properties for bone regeneration [28]. The development of bone scaffolds or cements has been widely achieved utilizing silk fibroin (SF) in the form of matrix or additives. In recent reports, SF assembled into nanofibers and was used in tuning the solidification of CS, which accomplished superior biological and mechanical performance, in comparison to that of natural and amorphous SF-blended CS [17]. The composite cements with natural SF nanofibers elicited a higher mechanical property, and, hence, it was suggested to have a favorable role in bone regeneration.

Techniques such as electrospinning and phase separation are commonly used to produce scaffolds but face the difficulty of complete solvent elimination [29]. Usually, SF is produced by the process of freeze drying [26,30], but according to Li et al., the CaCl_2_-ethanol solution may be regarded as an apt method for the preparation of SF as a biomaterial [31]. The silk fibroin is bio-fabricated by solution processing due to its thermal degradation of β-sheet nanocrystallites in the melting process. Many efforts have been made to dissolve silk fibroin using 9.3 M LiBr solution or a solvent mixture of CaCl_2_/EtOH/H_2_O to break the hydrogen bond network [32,33]. However, these approaches require downstream processing related to salt-containing solvent removal through time-consuming steps like dialysis and freeze drying. Although SF/CS composite synthesis and properties have been reported, there are no studies reporting the dialysis-free process of SF/CS composites. Hence, a novel strategy to synthesize CS and simultaneously produce the SF/CS composite is reported in this work. Furthermore, the synthesized SF/CS samples were characterized on the basis of their improvements in crystal structure and morphological, chemical, and mechanical properties.

## 2. Materials and Methods

### 2.1. Synthesis of SF/CS Composites by a One-Pot Process

The sericin from the native silkworm cocoons was removed by a degumming process by autoclaving the silkworm cocoons for 1 h. Then, the degummed fibers were dissolved in a solvent mixture of CaCl_2_/C_2_H_5_OH/H_2_O (1:2:8) to prepare a 5 wt.% SF solution and centrifuged at 6000× *g* rpm for 5 min. All the reagents were procured from Sigma Aldrich, New Taipei City, Taiwan and were of >98% purity. Using a one-pot process, the designed stoichiometric amount of Na_2_SO_4_ was then individually added to the SF solution at 90 °C for 2 h, then rinsed with 60% ethanol for 30 min, and oven dried overnight, thereby forming SF/CaSO_4_, with compositions as listed in Table 1.

### 2.2. Characterization of the SF/CS Composites

The SF/CaSO_4_ powders were harvested and examined for the crystalline pattern by X-ray diffraction (XRD, D/MAX-RC, Rigaku, Tokyo, Japan) with a Ni filter and CuKα radiation (λ = 0.154 nm) at 30 kV and 20 mA. Measurements were conducted in a continuous scan mode with a scanning rate of 10°/min and 2θ from 5° to 60°. Differential scanning calorimetry (DSC) is a convenient method to determine the miscibility of polymer blends. For the thermal analysis, the DSC was performed on the samples of 5 mg, which were loaded in aluminum DSC pans, hermetically sealed with a crimping press, and placed in a DSC oven (TA Instrument, TA Q100, New Castle, DE, USA). The temperature was elevated from room temperature to 300 °C at a rate of 10 °C/min. The chemical composition of the SF/CaSO_4_ powders was characterized using the Fourier Transforms Infrared Spectroscopy (FTIR) spectrometer (Tensor 27, Bruker, Madison, WI, USA). The infrared spectra with a resolution of 4 cm^−1^ under 16 scans were adopted with a scanning range of 400–4000 cm^−1^. The microstructural characterization of the crystal morphology of all the sputter-coated samples with an ion sputter (S-3000H, Hitachi, Tokyo, Japan) was examined by scanning electron microscopy (SEM, SU3500, Hitachi Ltd., Tokyo, Japan) at 15 kV and a magnification of 1000×.

After mixing the SF/CF cement powder with DDW, using a P/L ratio of 3:1, pastes were loaded into a cylindrical Teflon mold (6-mm diameter and 5-mm height) and allowed to set for 1 day. The compressive test was preformed using universal test machine (LF plus, AMETEK Co., Largo, FL, USA) at a crosshead speed of 0.5 mm/min.

### 2.3. Data Analysis

The one-way ANOVA test was used to evaluate the statistical significance of the measured results. When the analysis indicated significant variances between group means, each group was compared using Turkey’s multiple comparison test. The results of *p* < 0.05 were considered statistically significant.

## 3. Results and Discussions

### 3.1. The Crystalline Phase Analysis of SF/CS by X-ray Diffraction

The CaCl_2_-based solvent system has been commonly used to dissolve silk fibroin [32,33]. Unlike the typical dialysis process for salt removal, in this study, calcium chloride was used as the raw material to react with sodium sulfate to form an effective dispersion of calcium sulfate in silk fibroin, inducing the precipitation of SF/CS composites through a one-pot reaction. The crystal pattern of pure CS and the harvested SF/CS samples subjected to 60% ethanol treatment was examined by the XRD technique and is represented in Figure 1. While the XRD peaks at 11.64° (020), 20.75° (021), 23.41° (040), and 29.14° (041) correspond to crystal patterns of calcium sulfate dihydrate (CSD), the diffraction peaks at 14.76° (110), 25.67° (310), 29.77° (220), and 31.91° (−114) indicate the crystal patterns of CSH [17]. Unlike the pure CS sample, which reveals the character of CSD, the SF/CS composite groups display typical diffraction peaks similar to those of CSH. The effect of adding SF (10–50%) to CS maintains the desired CSH form and restricts the formation of CSD. Consequently, the characteristic peaks of CSH (50–90% CS in SF/CS composites) and CSD (100% CS) appear sharp and clear, and they indicate almost no (or only small peaks) unreacted materials or impurities.

In a previous study, upon SF solution (6 g/L) being mixed with CSH powder (ratio of 0.40 mL/g), the SF/CS material was found to exhibit the dehydrated form, as seen through XRD analysis [32]. Furthermore, it was noted that the CSH form was converted to CSD while preparing composite SF/CS samples [17]. In contrast to the aforementioned results, the SF/CS composites synthesized in this study gave rise to the formation of a CSH phase, despite being mixed with the SF.

### 3.2. DSC Thermal Analysis of SF/CS Composites

As the XRD analysis was unable to verify the presence of the α or β form, the samples were then characterized by DSC. The first DSC heating curves (Figure 2a) show two different endothermic peaks at ~151 and ~195 °C, demonstrating the characteristics of CSD [34]. With increasing silk fibroin content, the endothermic peak intensity reduces at first and then appears prominent at (50/50), which indicates the possible presence of CSD (although with shifted peaks). The peak at ~195 °C disappears after the addition of silk fibroin, indicating the probable absence of calcium sulfate hemihydrate, with CSD formation unable to be clearly verified. Therefore, DSC analysis cannot lead to a definite conclusion about the forms of as-synthesized calcium sulfate.

Similar to the miscible polymer blend, SF/CS composites display a single glass transition temperature (Tg) and shift to a higher temperature with increasing CS content, indicating the possible molecular-level interaction between CS and SF. The SF and CS in the composite are assumed to be occurring in two different phases without actually reacting. However, Figure 2b shows a single Tg in the DSC heating curve for each SF-based composite. The Tg of the hybrid composites is found to be progressively moving towards higher temperatures as the CS component of the hybrid increases. This may be attributed to the reduced segmental mobility of SF in the samples due to the increased incorporation of the hard and compact CS moiety on the thermal properties of the SF polymer, hence enhancing the thermal stabilities of the organic–inorganic composites. This increase in the Tg value may also be indicative of a possible molecular lever interaction between the SF polymer and CS ceramic. Recognition at the molecular level of the inorganic–organic interface has been found to control crystallization and mineralization in synthetic and biological systems [35]. This might be related to the existence of CSH and its role in interacting with SF for a similar kind of phenomenon, as discussed previously.

### 3.3. FT-IR Analysis of SF/CS Composites

To elucidate the molecular mechanism that the hydrothermal process had exerted upon the spider silk threads, FT-IR analysis was performed. The spectra (Figure 3a) shows that the 3350–3650 cm^−1^ peaks correspond to the O–H vibrations of water molecules in HH crystals. Particularly, the peaks at 1100–1200 cm^−1^ are related to the asymmetric stretching of ν_3_ SO_4_, and the peak at 660 cm^−1^ is related to the stretching of ν_4_ SO_4_ [36]. The 1622 and 1550 cm^−1^ peaks are responsible for the C=O and N-H groups, respectively. This may be attributed to the interaction that SF administration brought into the composite. In addition, the peak intensity corresponding to SO_4_ decreases at first (SF content increases) and then diminishes (100% SF only), indicating the successful formation of the SF/CS composite, and some peaks are slightly displaced or may be weakened by the superposition. However, no new peaks appear which indicates the possibility of SD and CF combining and/or interacting individually rather than chemically reacting. 

Peak deconvolution was conducted using PeakFit 4.11 for the major functional peaks of the β-sheet and random coil in order to demonstrate the structural transition occurring (Figure 3b). The SF/CS (100/0) composite shows the peak corresponding to the β-sheet at 1622 cm^−1^ and random coil at 1656 cm^−1^, which slightly shifts with the addition of CS. The ratio of the area under the β-sheet and random coil is depicted in Table 2, with the maximum ratios of 1.77 and 1.86 for SF/CS (50/50) and (20/80) samples, respectively.

It was observed that when SF was prepared through a water-based solution at a neutral pH and room temperature, it formed α-helix and random coils, which possess poor mechanical strength [37]. In order to enhance the strength of SF by the formation of β-sheet structure, various methods have been introduced, such as chemical, physical and enzymatic crosslinking, resulting in conformational transitions to the β-sheet from the random coil [38,39]. Moreover, adding salts, surfactants, or metal ions quickens the structural transition in the SF aqueous solution and reduces its gelation rate [40]. Similarly, the addition of calcium sulfate to the SF enhances the β-sheet content, which may explain the possible increase in the mechanical strength, as observed later in this study.

### 3.4. Microstructural Properties Examined by SEM

The CaCl_2_/C_2_H_5_OH/H_2_O solvent, mixed with Na_2_SO_4_ and C_2_H_5_OH, may act similar to the crystal-controlling agent that causes dihydrate or hemihydrate forms. It can be seen that the SF/CS composite at the 0:100 ratio (Figure 4a) primarily exists in plate-like morphology, demonstrating the CSD form (however, traces of rod-like morphology can also be seen) [41]. Further, upon increasing the SF content, rod-like morphology of the CSH was observed in addition to the fibers of the SF embedded into the matrix (Figure 4b–f). The α-HH crystal form has developed transparent idiomorphic crystals, with sharp crystal edges, and the β-HH form has a flaky particle-like appearance, with small crystals [42]. Therefore, the morphology of the CSH in Figure 4 shows the appearance of the α form with clear needle-like edges. The β-HH is a softer material, with a short setting time and higher porosity, while the α form has smoother and denser particles. The longer setting time and lower porosity of the α-CSH lead to higher values of compressive and tensile strength after setting. Hence, alpha-hemihydrate has slow solubility and resorption [43]. It was reported that the loading of the *α*-HH bone morphogenetic protein-2 (peptide) repaired defects in the rabbit femoral condyle [44]. Moreover, it is the rapid solidification that makes *α*-CSH a candidate for bone cement. It was also observed that upon mixing *α*-CaSO_4_ 0.5H_2_O with bioactive glasses, the material demonstrated potential use as a bone implant substitute [45].

The two hemihydrate forms are different in terms of their water reactivity and hydration product strength [46]. We found no structural differences through XRD studies. Furthermore, no significant differences between the α and β forms were seen through infrared spectral studies. It was reported that the monoclinic and trigonal forms were crystallized from the α- and β-hemihydrate forms, respectively [46]. However, there were no crystallographic differences between the two forms, only changes in their size and crystal arrangement [46].

In our study, we conclude that there is formation of α-CSH upon introducing the SF content, and α-CSH is reported to have superior mechanical strength when compared to that of β form for biomedical applications [13].

### 3.5. Compressive Strength of the SF/CS Composites

The compressive strength of hydrated SF/CS cement composites was considerably altered with the increasing SF content, as shown in Figure 5. It was observed that the sample with 100% CS resulted in the lowest compressive strength, which was enhanced by increasing SF content by 10%, and was found to be maximum at 20% SF and 80% CS content (no significant difference when compared to SF/CS (10/90).

The SF protein is composed of hydroxyl and carboxyl groups, which chemically react with sulfate, perhaps leading to an enhancement of cohesion between SF and CS molecules. It has been reported that SF/CS/calcium phosphate (CP) cements and SF/CP improve mechanical properties [47], with another study showing that adding SF to calcium phosphate cement improves its properties [48]. In the same line as this study, a previous study also depicted the enhanced mechanical property of SF/CS bone cement, which showed further improvement by the addition of Sema3A-loaded chitosan microspheres [49]. In this study, the change in compressive strength is based upon the interaction between SF and CSH. As reported, there appeared to be an existing attraction between hydroxyl or carboxyl ends of the SF protein with the Ca^2+^ from CS, hence forming recrystallized CSH (earlier) whiskers with fiber-reinforced SF [17]. Upon further increasing the SF content, there was a decrement in the compressive strength. This decrement could be plausibly attributed to the agglomeration of the SF, causing a stress concentration issue [17]. Considering this, we speculate that the agglomerated SF, which likely restricts the effect of induced β-sheet domains leads to a non-uniform distribution, causing a reduction in the compressive strength beyond 20% SF + 80% CS while hardly influencing the content of the β-sheet in hydrogel [50]. It can also be assumed that the β-sheet content is not the only criteria contributing to the enhancement of the compressive properties, as there may be several other factors [50].

It was seen when the liquid-to-powder (L/P) ratio was varied, enhancement in the compressive strength was found to be higher. On the other hand, in our study, the L/P ratio was kept constant while having a large variation in the SF/CS ratios, which may have led to relatively lower values of compressive strength as reported [33]. However, the enhanced compressive strength of the SF/CSH composites in comparison to that of the pure CSD sample followed the same trend as that specified in the literature [17]. The compressive strength reported in this study falls in the same range as stated when CSH was added to SF/calcium phosphate (SF/CP). However, there was a decrement in the compressive strength of the SF/CP upon addition of CSH [47]. Based on these observations, we assume that the as-synthesized SF/CS composites lead to the increment in the compressive strength, which is approximately the same as that of the reported SF/CP composites [47]. This clarifies the efficacy of SF in enhancing the mechanical properties of CSH up to a desired limit.

The biocompatibility of materials represents one of the most important features for their use as implants in humans. CS possesses good biocompatibility with bone conduction and degradation properties [51]. The mechanisms by which CS enhances bone formation have not been completely elucidated. CS likely releases ionic components, which can alter intracellular enzyme activity [52]. SF materials or SF-based cements support the adhesion, proliferation, and differentiation of cells and have no cytotoxic or immune response [53,54]. Biocompatibility analysis is required for the as-synthesized SF/CS materials in future work.

## 4. Conclusions

The one-pot wet precipitation process was found to be feasible to synthesize silk fibroin/calcium sulfate hemihydrate (SF/α-CSH) composites. The successful synthesis of the SF/CS composites was determined by the characterizations through XRD. The DSC characterization depicted interactions between the SF and CS at the molecular level. The FTIR analysis also confirmed the formation of SF/CS through the obtained functional groups. Furthermore, the SEM observations showed the formation of plated CSD and rod-like α-CSH embedding SF. The (10/90) and (20/80) SF/CS composites showed significantly higher compressive strength than that of pure calcium sulfate cement and other combinations of SF/CS. Biocompatibility studies are needed in order to conclude the clinical effectiveness of SF/CS in bone tissue engineering.

## Figures and Tables

**Figure 1 polymers-13-01996-f001:**
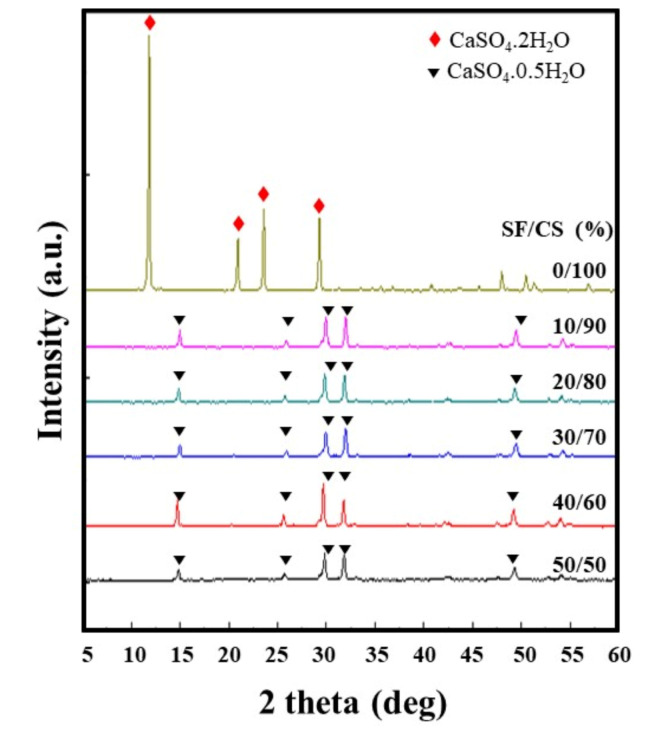
XRD pattern of various SF/CS composites.

**Figure 2 polymers-13-01996-f002:**
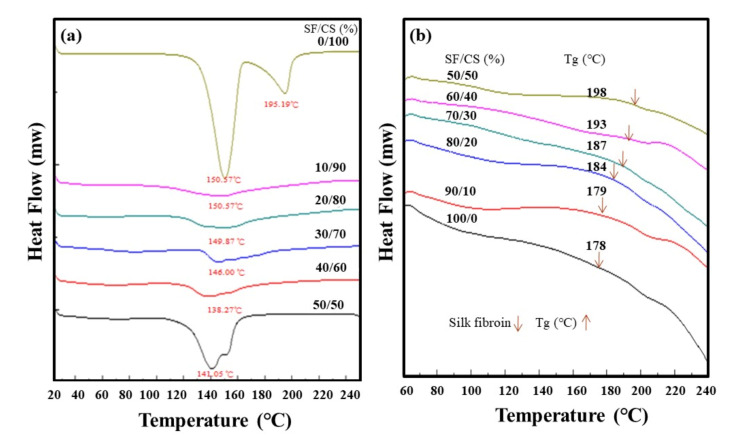
(**a**) DSC first heating curves of CS-dominating composites; (**b**) DSC second heating curves of SF-dominating composites.

**Figure 3 polymers-13-01996-f003:**
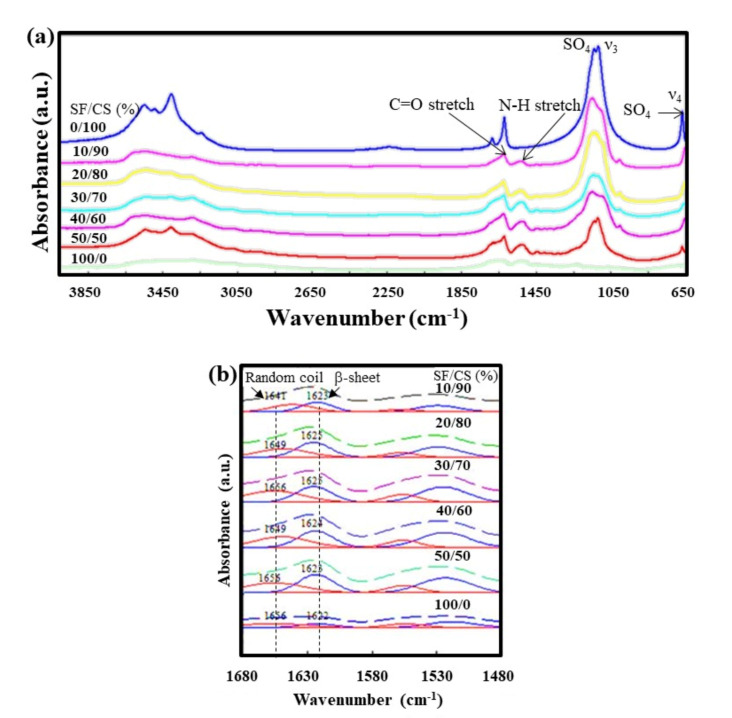
FTIR spectra of (**a**) the SF/CS composite; (**b**) its corresponding de-convoluted peaks showing random coil and β-sheet positions.

**Figure 4 polymers-13-01996-f004:**
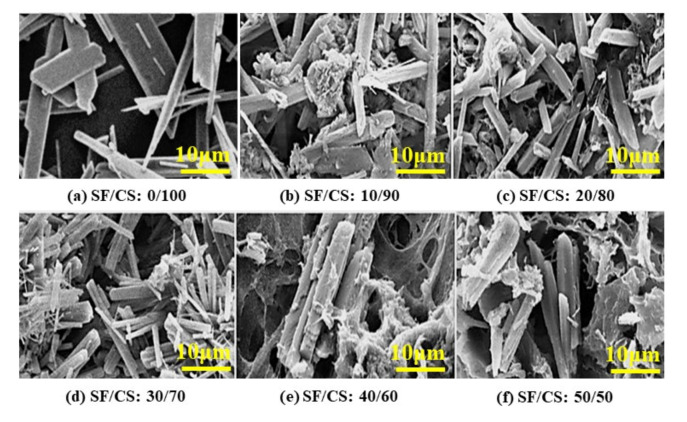
Representative SEM images of the various SF/CS composites, (**a**) SF/CS: 0/100, (**b**) SF/CS: 10/90, (**c**) SF/CS: 20/80, (**d**) SF/CS: 30/70, (**e**) SF/CS: 40/60, (**f**) SF/CS: 50/50.

**Figure 5 polymers-13-01996-f005:**
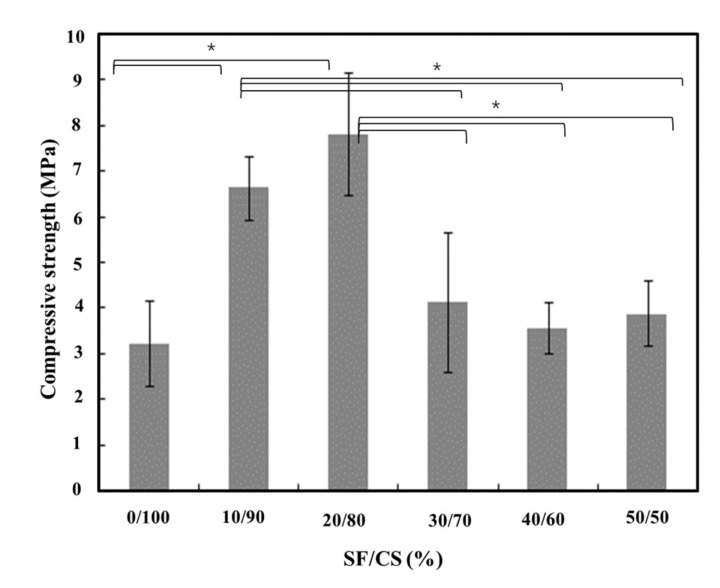
Compressive strength of various hydrated SF/CS cement composites (* represents *p* value < 0.05).

**Table 1 polymers-13-01996-t001:** Different components for SF/CS composites.

SF/CaSO_4_	0/100	10/90	20/80	30/70	40/60	50/50
Target SF conc. (%)	0	4	9	14	10	10
Na_2_SO_4_ (g)	41	39	37	35	16	10
CaSO_4_ (g)	39	38	36	34	15	10

**Table 2 polymers-13-01996-t002:** Estimation of ratio of β-crystal/random coil at various SF/CS ratios.

SF/CS (%)	β-Crystal	Random Coil	Ratio of β-Crystal/Random Coil
10/90	0.08	0.07	1.14
20/80	0.13	0.07	1.86
30/70	0.14	0.10	1.40
40/60	0.15	0.10	1.50
50/50	0.16	0.09	1.77
100/0	0.03	0.04	0.75

## Data Availability

Publicly available datasets were analyzed in this study. This data can be found here: https://hdl.handle.net/11296/ytr6fr.

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
