# Peer review of "A Novel One-Pot Synthesis and Characterization of Silk Fibroin/α-Calcium Sulfate Hemihydrate for Bone Regeneration"

_polymers, 2021, doi:10.3390/polym13121996_

Round 1
Reviewer 1 Report
The authors investigated the production of silk fibroin/calcium sulfate (SF/CS) composites by one-pot synthesis for bone regeneration applications. The manuscript can be accepted after considering the following comments:
- The abstract part is still unclear and must be rewritten, it has many typos and the chemical structures should be checked. The valence of each element should be subscripted
The degradation parts in DSC Images (Figure 2) are not clear, enhance them with more resolutions.
Scale bars of Figure 4 are missed, thus it is very difficult to compare the morphological structure of each sample.
The authors discussed most of the data in the present simple and the others in the past simple. please check and carefully revise the language of the current manuscript. Please recheck.
Figure 2 (a, b) should be redrawn with sharp color to be more readable for the readers.
Reviewer 2 Report
In the FTIR deconvolution, the peak assignments are still incorrect. The peak around 1623 cm-1 should be beta-sheet while the random coil/helix is around 1650 cm-1. Please revise.
Reviewer 3 Report
The manuscript “A Novel One-pot Synthesis and Characterization of Silk Fibroin/α-Calcium Sulfate Hemihydrate for Bone Regeneration” deals with the production of silk fibroin/calcium sulfate (SF/CS) composites by one-pot synthesis, for bone regeneration. Good results, in terms of chemical, thermal and morphological properties of the biomaterials, have been obtained. Moreover, the work is well organized and written. The publication is recommended after minor revisions, as follows:
- The state of the art on the use of silk fibroin in tissue engineering can be enlarged; see, for instance, this work: Loaded silk fibroin aerogel production by supercritical gel drying process for nanomedicine applications, Chemical Engineering Transactions, 2016, 49, pp. 343–348.
- References in the text are not in line with the Journal guidelines. Check and correct them.
- Remove number 18 between ref. 37 and 38.
Author Response
Please see attachment.

This manuscript is a resubmission of an earlier submission. The following is a list of the peer review reports and author responses from that submission.
Round 1
Reviewer 1 Report
The manuscript “A Novel One-pot Synthesis and Characterization of Silk Fibroin/a-Calcium Sulfate Hemihydrate for Bone Regeneration” deals with the production of bio-composites for bone tissue engineering. Mechanical properties and the general bio-composites performance were improved thanks to the addition of silk fibroin into the calcium sulfate matrix. The work is well organized and written; moreover, it is in line with the aims of the Journal. Therefore, the publication is recommended.
Detailed comments:
- Introduction. The state of the art related to the bone tissue engineering can be enlarged, focusing on the selection of the proper biopolymers for scaffold production. As an example, see this review: Baldino et al., Regeneration techniques for bone-To-Tendon and muscle-To-Tendon interfaces reconstruction, British Medical Bulletin, 2016, 117, pp. 25-37; etc..
- Results. SEM images reported in Figure 4 do not show a uniform morphology of the bio-composites; i.e., a 3-D porous matrix, generally required for this kind of application, is missing. Of course, this result also influences the mechanical properties of the composites. Since this is a relevant aspect, a more detailed discussion should be added to justify the results obtained and the suitability for bone regeneration.
- A more accurate comparison with the literature should be performed to highlight the relevance of the results found in this work.
Reviewer 2 Report
The overall quality of the work needs significant improvement. Here are some major comments for consideration:
- The language and spelling need serious revision.
- Some experimental details are missing. For example, how to remove NaCl in the system.
- For FTIR characterization, the deconvolution and peak assignments are incorrect.
- Since the authors claimed the materials are used for bone regeneration. Related biology studies (e.g., in vitro cell culture or in vivo) are needed.
Reviewer 3 Report
The authors investigated the production of silk fibroin/calcium sulfate (SF/CS) composites by one-pot synthesis for bone regeneration applications. The manuscript can be accpeted after considering the following comments:
- The abstract part must be rewritten, it has many typos and the chemical structures should be checked. The valence of each element should be subscripted.
- the characterization tools should be combined in one paragraph with no subtitles.
- In line 105, parenthesis should be deleted.
The first sentence in line 170 should be moved to the characterization part.
The degradation parts in DSC Images (Figure 2) are not clear, enhance them with more resolutions.
The numbers of X-axis (Figure 3 a) are intertwined with each other, redraw the figure to be more readable with no intertwined.
Scale bars of Figure 4 are missed, thus it is very difficult to compare the morphological structure of each sample.
The authors discussed most of the data in present simple and the others in past simple. please check and carefully revise the language of the current manuscript